# Identifying individuals with high risk of Alzheimer's disease using polygenic risk scores

Ganna Leonenko[1,6], Emily Baker [1,6], Joshua Stevenson-Hoare[1], Annerieke Sierksma [2,3], Mark Fiers [2,3,4], Julie Williams [1,5], Bart de Strooper [2,3,4] & Valentina Escott-Price [1,5✉]

Polygenic Risk Scores (PRS) for AD offer unique possibilities for reliable identification of individuals at high and low risk of AD. However, there is little agreement in the field as to what approach should be used for genetic risk score calculations, how to model the effect of *APOE*, what the optimal *p*-value threshold (pT) for SNP selection is and how to compare scores between studies and methods. We show that the best prediction accuracy is achieved with a model with two predictors (*APOE* and PRS excluding *APOE* region) with pT<0.1 for SNP selection. Prediction accuracy in a sample across different PRS approaches is similar, but individuals' scores and their associated ranking differ. We show that standardising PRS against the population mean, as opposed to the sample mean, makes the individuals' scores comparable between studies. Our work highlights the best strategies for polygenic profiling when assessing individuals for AD risk.

[1] UK Dementia Research Institute, Cardiff University, Cardiff, UK. [2] VIB Center for Brain & Disease Research, Leuven, Belgium. [3] Laboratory for the Research of Neurodegenerative Diseases, Department of Neurosciences, Leuven Brain Institute (LBI), KU Leuven (University of Leuven), Leuven, Belgium. [4] UK Dementia Research Institute, University College London, London, UK. [5] Division of Psychological Medicine and Clinical Neurosciences, School of Medicine, Cardiff University, Cardiff, UK. [6] These authors contributed equally: Ganna Leonenko, Emily Baker. ✉email: escottpricev@cardiff.ac.uk

Alzheimer's disease (AD) is the most common type of dementia, and mainly affects the elderly population. AD is a progressive condition, which means that clinical features develop gradually over many years before diagnosis[1]. The ability to predict AD risk before disease onset is of great importance for stratifying people for clinical trials or the selection of candidates for functional experimental studies. Given the time and labour costs associated with these objectives, the selection of those individuals at high or low risk of developing AD must be as reliable as possible.

About 35% of life-time risk of dementia is modifiable by factors such as education, nutrition, health care, and social deprivation[2], including a better management of vascular risk factors with their prevalence decreasing over time. The effects of better management of these risk factors manifests in a delay in age at onset of the disease as noticed in many studies[3–5]. However, with increased lifespan the prevalence of the disease still goes up[6]. If controls are enrolled from the population and/or are younger than cases, then a proportion of them will develop AD at a later time[7]. Due to potential delay of the age at onset, even age-matched control samples are likely to encompass future AD cases who are yet to show symptoms. Therefore, genes with small effect sizes associated with AD due to age-related pathological changes[8] can be overlooked in such studies. For example, the *APOE-ε4* allele is associated with earlier age at onset[9] but ε4 allele frequency in the population decreases from 0.18 to 0.09 with increasing age[10]. In a number of genome-wide association studies (GWAS), clinically defined AD cases are compared to cognitively normal individuals that are not necessarily age matched[11]. A recent study comparing AD cases with relatively young age at onset to centenarian-controls[12] observed that the effect sizes of GWAS significant SNPs in this study were on average twice as high as those calculated by the original GWAS studies, which confirms the importance of controls being age-matched to, or even older than, cases for prediction of AD risk.

Polygenic risk score (PRS) is used as a global term for a risk score including any number of SNPs. With the exception of *APOE-ε4*[13], the common genome-wide significant variants which have been discovered though GWAS have only small individual effects. Although it is clear that many genes are involved in disease development and progression, there is no agreement in the field as to whether AD is a polygenic or oligogenic disorder. Despite substantial evidence suggesting that the risk of AD is polygenic[14,15], more recent studies have argued in favour of an oligogenic view of AD[16,17]. To date, there is no consensus in the field about which PRS approach to use, how to model *APOE* within PRS, which *p*-value threshold for SNP selection is optimal or how to compare scores with other datasets when selecting individuals based on PRS.

When choosing individuals with more than 2 standard deviations (SD) from the PRS mean (i.e., PRS extremes)[18], the accuracy of distinguishing between individuals at high and low AD risk is high[18]. The choice of PRS calculation methodology may lead to identification of different sets of individuals even at the extreme ends of the resulting PRS distributions. All methods for PRS calculation attempt to reduce the signal to noise ratio by reducing the number of overall SNPs while keeping the most informative ones; of these methods, PRS with clumping and thresholding (C + T) is the simplest method[19]. Bayesian-based methods, by contrast, use all SNPs and offer strategies to adjust the effect sizes for LD instead of LD-clumping[20–23]. Functionally informed Bayesian approaches vary the strength of LD-adjustment for each SNP based on its functional annotations.

Thus, we sought to investigate the various methodologies and SNP selection approaches for AD risk prediction, with the aim to robustly predict at least those at very high or very low risk. First, to explain the discrepant views about the disease architecture (oligogenic vs polygenic), we investigated whether the disagreement about the optimal *p*-value threshold may arise from the unaccounted-for age-dependent frequency of *APOE-ε4*. This led us to look into how best to include the effect of *APOE* in PRS models, and what is the optimal *p*-value threshold (pT) for SNP inclusion in the PRS. Next, we compared a range of PRS calculation approaches including PRS(C + T) and Bayesian methods. Finally, we compared two PRS standardisation approaches: (a) standardisation of risk scores based on the cohort itself and (b) standardising against a population. Our conclusions are based upon examination of the number of and overlap between, individuals at high and low AD risk, identified by the different PRS calculation approaches and prediction models. To summarise, we provide recommendations about best practice for the robust identification of individuals at risk for AD using PRS.

## Results

**Optimal *p*-value threshold.** Table 1 details the description for each of the PRS models used throughout this manuscript. In our earlier work on PRS in AD[14,24] we have observed that using the directly genotyped *APOE* isoforms ε2 and ε4 as separate terms in the regression model in addition to the PRS excluding the *APOE* region (PRS.AD) provides higher prediction accuracy than modelling the *APOE* region as part of a full PRS. For this study we combine and harmonise ADNI, ROSMAP, MSBB, and MAYO datasets and use it as one case-control dataset (271 AD cases and 278 controls) in the manuscript, see details in Supplementary Tables 1 and 2. In the case-control dataset presented here, we observed that the optimal *p*-value threshold for the PRS depends upon how the *APOE* effect is accounted for. Table 2 presents the Area Under the Curve (AUC) and the variance explained ($R^2$) in the case-control dataset in three scenarios with four SNP *p*-value thresholds (pT ≤ 5e-8, 1e-5, 0.1, 0.5). The first section of the table shows the model with PRS calculated using the whole genome (PRS.full). In the second section the PRS was calculated excluding the *APOE* region (PRS.no.APOE). The third section shows the model with two independent variables i.e., PRS.no.APOE and *APOE*(ε2 + ε4) (PRS.AD).

The best prediction accuracy for the PRS.full model is achieved using genome-wide significant SNPs, pT ≤ 5e-8, (AUC = 69.8%), but this is not better than *APOE*(ε2 + ε4) alone (AUC = 70.0%). When more risk genes are included by relaxing the *p*-value

**Table 1 Model description for the PRS models presented in the manuscript.**

| Model Name | Model description |
|---|---|
| ORS.full | PRS including SNPs with a pT ≤ 1e-5 |
| ORS.no.APOE | PRS including SNPs with a pT ≤ 1e-5 and excluding SNPs in the *APOE* region (chr19:44.4-46.5 Mb) |
| PRS.full | PRS including SNPs with a pT ≤ 0.1 (unless otherwise specified) |
| PRS.no.APOE | PRS including SNPs with a pT ≤ 0.1 and excluding SNPs in the *APOE* region (chr19:44.4-46.5 Mb) (unless otherwise specified) |
| PRS.AD | PRS calculated as a weighted sum of PRS.no.APOE (including SNPs with a pT ≤ 0.1, unless otherwise specified) and *APOE*(ε2 + ε4), where *APOE* effects were weighted with effect sizes (B(ε2) = −0.47 and B(ε4) = 1.12) as in Kunkle et al. 2019 |

**Table 2 PRS prediction accuracy for the AD case-control dataset using different *p*-value thresholds and methods to model *APOE*.**

| pT | PRS.full | | | | PRS.no.APOE | | | | PRS.AD | | |
|---|---|---|---|---|---|---|---|---|---|---|---|
| | N SNPs | AUC (%) | $R^2$ | OR (95% CI) | N SNPs | AUC (%) | $R^2$ | OR (95% CI) | AUC (%) | $R^2$ | OR (95% CI) |
| *APOE* (ε2 + ε4) | 2 | 70.0 | 0.18 | 2.2 (1.8,2.7) | – | – | – | – | 70.0 | 0.18 | 2.2 (1.8, 2.7) |
| 5e-8 | 65 | 69.8 | 0.16 | 2.2 (1.8, 2.7) | 17 | 55.7 | 0.02 | 1.2 (1.0, 1.5) | 71.4 | 0.19 | 2.4 (2.0, 3.0) |
| 1e-5 (ORS) | 126 | 69.4 | 0.16 | 2.2 (1.8, 2.7) | 66 | 56.7 | 0.02 | 1.2 (1.1, 1.5) | 72.0 | 0.20 | 2.4 (2.0, 3.0) |
| 0.1 | 68,681 | 64.9 | 0.09 | 1.8 (1.5, 2.2) | 68,516 | 61.3 | 0.06 | 1.6 (1.3, 1.9) | 74.1 | 0.24 | 2.8 (2.2, 3.4) |
| 0.5 | 203,950 | 62.6 | 0.07 | 1.7 (1.4, 2.0) | 203,710 | 60.5 | 0.05 | 1.5 (1.3, 1.8) | 73.7 | 0.23 | 2.7 (2.2, 3.4) |

Legend: PRSs were calculated on a case-control cohort (271 clinically defined AD cases and 278 cognitively normal controls) using Kunkle et al. (2019) summary statistics for pT ≤ 5e-8, 1e-5, 0.1, 0.5 LD-pruned SNPs and APOE(ε2 + ε4). The number of SNPs (NSNPs) in each risk score are reported. Three PRS models were considered: PRS.full calculated on the full summary statistics; PRS.no.APOE where the APOE region was excluded (chr19:44.4-46.5 Mb); PRS.AD which is calculated as a weighted sum of PRS.no.APOE and APOE(ε2 + ε4), where APOE effects were weighted with effect sizes (B (ε2) = −0.47 and B(ε4) = 1.12) as in Kunkle et al (2019). The number of SNPs for PRS.AD models is always two more than for PRS.no.APOE. Prediction was estimated in terms of AUC, $R^2$ and OR with 95% Confidence Intervals (CI).

threshold the AUC decreases to 62.6% (first section of Table 2). The lowest prediction accuracy is observed with the PRS.no.APOE model excluding the *APOE* locus. The prediction accuracy does, however, increase from AUC = 55.7% for pT ≤ 5e-8 to 61.3% for pT ≤ 0.1. Note that the results do not change much between pT ≤ 0.1 and pT ≤ 0.5, despite the inclusion of 3 times as many SNPs at pT ≤ 0.5. The best prediction accuracy (AUC = 74.1%) and the highest variance explained ($R^2 = 0.24$) is achieved by the PRS.AD model where PRS.no.APOE is combined with *APOE*(ε2 + ε4) (last section of Table 2), using pT ≤ 0.1. The results of the PRS.full model, conversely, show a rather paradoxical trend that the prediction accuracy decreases when including more risk SNPs, i.e., by relaxing the pT threshold. The results above are based on PRS with LD-clumping parameter $r^2 > 0.1$. Similar patterns were observed when we used $r^2 > 0.01$ and $r^2 > 0.001$, however, the total prediction accuracy was slightly reduced (Supplementary Table 3). These opposing results with different *p*-value thresholds reflect very well the current controversies in the field. To investigate why such contradictory conclusions may be drawn from the same data, we set up a simulation study.

We make the assumption that the population controls are younger than cases for our simulations, as this is often observed in real studies. This implies that some of the control population have not reached the age of disease onset yet. Based upon ε4 frequency and studies of ε4 dependent age at onset[9], we estimate that 28% of them will develop AD. Accounting for the prevalence of cases[13] (34%), using reported *APOE*-ε4 allele frequency in the whole sample of 0.216 and OR = 3.326, we calculate the allele frequencies in cases and controls as 0.356 and 0.142, respectively. Then we simulated 67 SNPs with effect sizes and allele frequencies[13] corresponding to SNPs with pT ≤ 1e-5, along with 10,000 SNPs with a range of allele frequencies (0.01 to 0.45) and effect sizes decreasing from OR = 1.005 to 1. We calculated the Oligogenic Risk Score (ORS.full) based on 68 SNPs (including *APOE*-ε4) with pT ≤ 1e-5, PRS.full based upon 10,068 SNPs, and PRS.AD as PRS.no.APOE combined with a separate variable *APOE*-ε4 (see Table 1 and Methods section for details). The comparison of the prediction accuracy by the ORS.full, PRS.full, PRS.AD has shown the contradictory pattern of AD risk prediction, similar to that observed in other AD PRS studies[16,17] (Supplementary Fig. 1). In particular, ORS.full has an advantage over PRS.full, however when *APOE* is accounted for separately in addition to PRS.no.APOE, PRS.AD has the best prediction accuracy (AUC) and variance explained ($R^2$).

Informed by the simulation results, we explored the *APOE*-ε4 allele frequencies in the case-control dataset with age (see Fig. 1A). As reported in other studies, the ε4 allele frequency in this data set decreases with age, the ε3 frequency increases and ε2

frequency remains approximately the same. Figure 1B, C shows that ε4 frequency reduces faster in cases than in controls (red line). The oligogenic risk score, ORS.no.APOE (based on SNPs with pT ≤ $10^{-5}$), also decreases with age in cases but is on average higher than in controls, with the highest being in ORS.no.APOE for ε44 cases as reported in[17]. Contrary to ORS.no.APOE, the mean of PRS.no.APOE (blue line) is higher in older cases and lower in older controls[25]. Thus, because of the changing allelic frequencies of *APOE* genotypes over age, it is clear that both the *APOE* genotype by itself and ORS.no.APOE become much less accurate predictors in older cases, while the reverse is seen with PRS.no.APOE. Clearly, *APOE* and ORS will serve as better predictors of AD risk at younger ages. Here we find that PRS increases with age, but whether this is a true effect or is due to random variation requires further investigation and replication. Figure 1 shows that the net age effect for the sum of ORS and PRS is smaller than the separate score changes with age. Since these changes are in opposite directions, they cancel each other out if taken as a sum. Moreover, the net effect is approximately the same in cases and in controls. This net effect corresponds to the model that is referred to as polygenic in the field and leads to conclusions in favour of an oligogenic model. However, the differential age effect, leveraging the polygenic disease architecture, can only be discovered when considering *APOE* (and/or ORS) and PRS.no.APOE separately. Adjusting the combined score for age only corrects for the small net effect. Thus, these sample and simulation data demonstrate that even though the ORS is a good predictor for AD at younger ages, it is mainly driven by the age-specific *APOE* allele frequency distribution.

**Comparison of PRS calculation approaches.** Until now, we have solely used the PRS(C + T) method for the calculation of PRS. Calculation methods of PRS are based on different assumptions, and an important consideration is what are the most reliable methods to predict the right patients versus controls with maximal accuracy. PRSice[26] is a software which implements the PRS (C + T) method automatically and so the same LD-clumping parameters were specified for this approach. LDAK[20] does not require LD-clumping and calculates PRS adjusting SNP effect sizes for LD by reducing the contribution of SNPs in regions of high LD. LDpred-inf[21], PRS-CS[22] and SBayesR[23] are all Bayesian approaches which use estimates of SNP effect sizes based on SNP-based heritability and also account for regional LD structure. Figure 2 shows the results of prediction accuracy of ORS and PRS for six different methods of PRS calculation, namely PRS(C + T), PRSice, LDpred-inf, PRS-CS, LDAK and SBayesR. The highest prediction accuracy was found in our case-control sample for both ORS.full and PRS.full using PRS(C + T) with AUC =

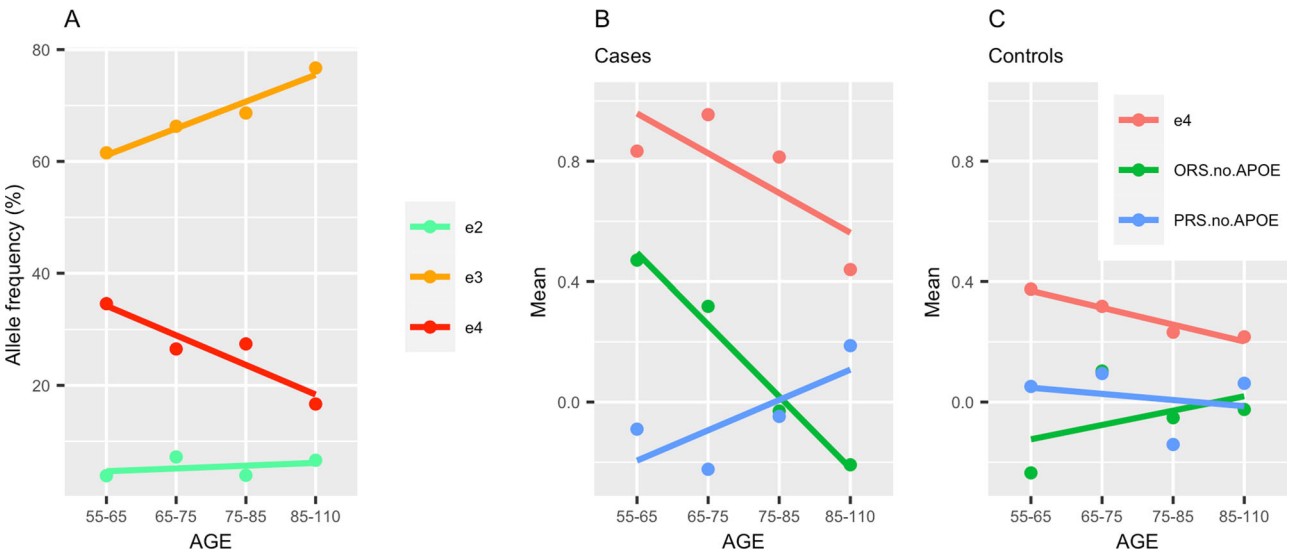

**Fig. 1 Effects of APOE allele frequencies and age on genetic risk scores. A** Allele ε4, ε3, ε2 frequencies (red, orange and green lines respectively) in the case-control dataset (271 cases and 278 controls), **B**, **C** the mean risk score for ε4 allele frequencies, ORS.no.APOE and PRS.no.APOE (red, green and blue lines respectively) split by age groups. ORS.no.APOE includes SNPs with $p$-value ≤ 1e-5, PRS.no.APOE includes SNPs with $p$-value ≤ 0.1 and both exclude the APOE region. **A** represents the full sample (**B**) represents cases only and (**C**) controls only. Age groups are specified as (55–65, 65–75, 75–85, 85+). For comparability of the scores in (**B**, **C**) the e4 genotypes (originally coded as 0/1/2) were also standardised. ORS Oligogenic risk score, PRS Polygenic risk score, SNP Single nucleotide polymorphism.

65–70% ($R^2 = 0.09$–$0.016$), and lowest for SBayesR with AUC = 54–61% ($R^2 = 0.01$–$0.05$). It should be noted that LDpred-inf, PRS-CS, LDAK and SBayesR do not require $p$-value thresholding. Therefore, we computed PRS in the full SNP set for LDpred-inf, PRS-CS and LDAK (SBayesR would not run for all chromosomes for all SNPs), and results were similar to those from PRS using thresholded SNPs with pT ≤ 0.1 (AUC = 59.3%, 69.6% and 59.7% respectively).

PRS(C + T) and PRSice showed very similar results across all prediction metrics, which is anticipated as both methods use the same approach, with PRSice performing an automatic filtering of SNPs that may differ from PRS(C + T). We also computed the PRS.AD model using each method. In line with the earlier conclusions, both prediction metrics (AUC, $R^2$) are better when APOE is modelled separately and subsequently added to the PRS. no.APOE for all methods (AUC = 73–74%, $R^2 = 0.22$–$0.24$). The detailed results can be seen in Supplementary Table 4.

**Population-based standardisation.** We compared AD ORS and PRS distributions, the latter with and without APOE calculated with the PRS(C + T) approach in two European populations; UKBB ($N = 364{,}236$) and 1000 Genomes ($N = 503$). Both populations are European, however they vary by sample size and genotyping platform. When comparing PRS(C + T) distributions for 1000 Genomes and UKBB, it can be observed that the two distributions are very similar at $p$-value thresholds of pT ≤ 5e-8, 1e-5 and 0.1, see Supplementary Fig. 2. More differences can be observed though at pT ≤ 0.5, where the UKBB PRS distribution has its mean slightly shifted to the left and a smaller standard deviation than that of the 1000 Genomes. The shift of the mean can be explained by the fact that UKBB participants report fewer illnesses, higher education and occupation than the UK general population[27], which are known to modify life-time risk of AD[2]. The smaller SD of the single-country UKBB-PRS (based on large number of SNPs) is also expected when compared to a sample comprising individuals from a number of European countries (1000 Genomes). For SNPs with an AD risk association $p$-value below the threshold (pT ≤ 0.1) the AD PRS distribution

parameters are sufficiently similar, and for ease-of-use reasons we therefore decided to work with the 1000 Genomes hereafter.

When comparing the PRS.AD distributions of the case-control dataset standardised (a) within the dataset and (b) against 1000 Genomes (Supplementary Fig. 3, Supplementary Table 5), it can be clearly seen that, as expected, the PRS distribution of the population lies between controls (shifted to the left) and cases (shifted to the right). In addition, the population-based standardisation increases the variation in the case-control sample, leading to more cases and controls falling above and below a predefined PRS cut-off (e.g., 2 SD), respectively.

**Individuals at the extreme tails of the PRS distribution.** We next investigated to what extent the PRS score can be used to identify, with good confidence, individuals with high and low risk of AD. We define PRS extremes as individuals with a score exceeding ±2 SD from the data mean or from the population mean, depending on the method of standardisation. We assess the effects of 1000G-based standardization on a human iPSC resource, HipSci, which is population based, as well as on a case-control dataset. For the PRS.AD model, when the HipSci sample is standardised within the sample 11 positive and 2 negative extremes are observed. When standardised against the 1000 G population cohort there are 6 positive and 5 negative extremes. It appears that standardisation of the HipSci data against the population provides no advantage above considering them internally as the PRS distributions in the population and in the population based HipSci should be the same.

In a case-control dataset the number of positive and negative extremes is greater when PRS is standardised against the population than within the sample (see Table 3 and Supplementary Table 4). The highest OR and prediction accuracy is observed with PRS.AD (OR = 124, AUC = 88.2) and the lowest with ORS. full (OR = 10, AUC = 74.6). Often, when selecting individuals at the extremes of risk for AD, researchers may want to understand risk beyond APOE. Thus, in Table 3 we also present the results for extremes selection in APOE-ε3 homozygotes using a score excluding the APOE region. As expected, the number of extremes

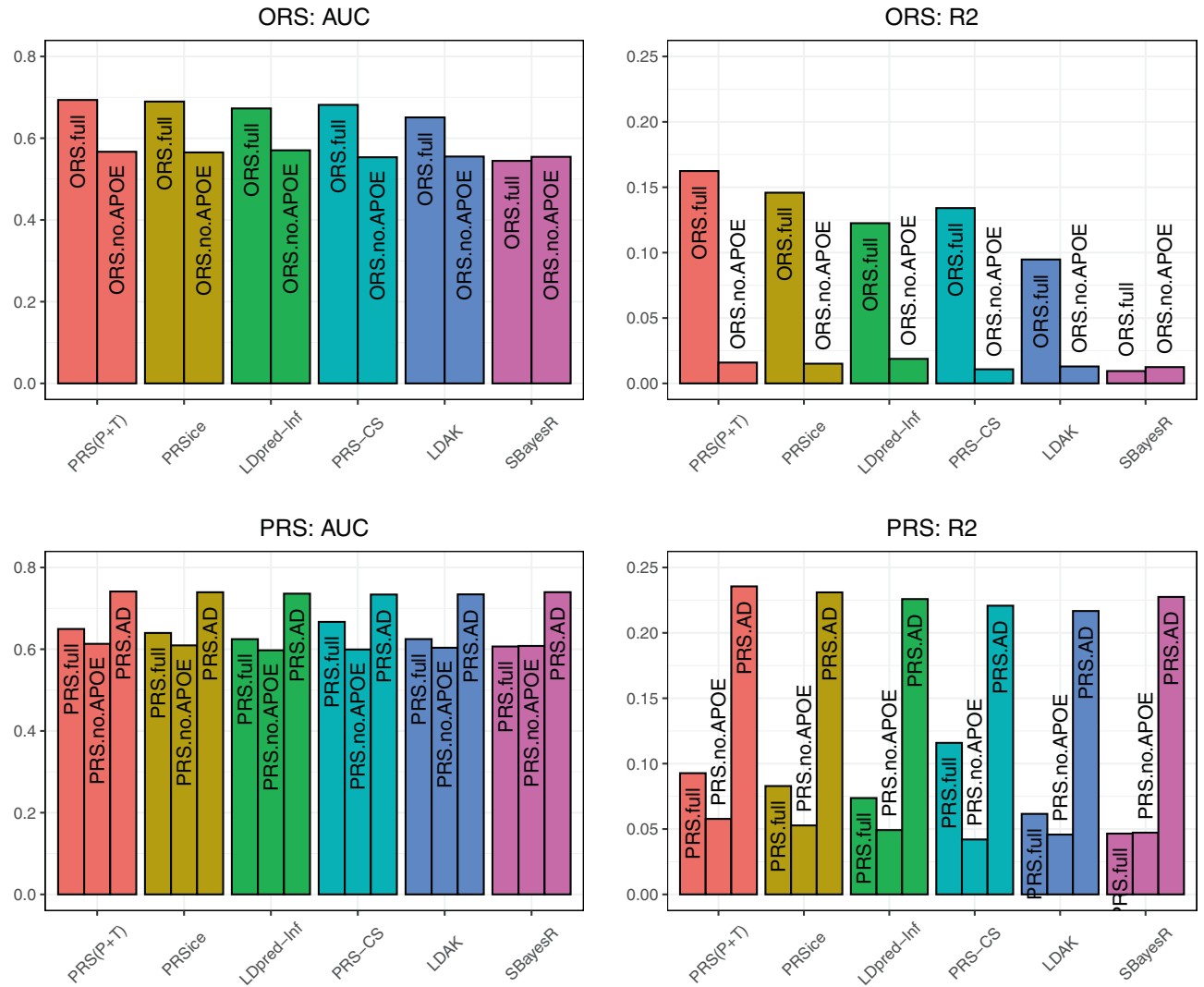

**Fig. 2 Prediction accuracy across different PRS methods (PRS(C + T), PRSice, LDpred-Inf, PRS-CS, LDAK and SBayesR) for ORS.full, ORS.no.APOE, PRS.full, PRS.no.APOE and PRS.AD.** Bar plot for prediction accuracy (AUC and $R^2$) across 6 PRS approaches: PRS(C + T), PRSice, LDpred-Inf, PRS-CS, LDAK and SBayesR (red, yellow, green, teal, blue, pink bars respectively). The colour of each PRS method is consistent across all plots. Upper figures represent ORS and lower figures represent PRS and PRS.AD models in the case-control dataset (271 cases and 278 controls). ORS.full includes SNPs with pT $\leq$ 1e-5 and PRS.full includes SNPs with pT $\leq$ 0.1, ORS.no.APOE and PRS.no.APOE exclude SNPs in the *APOE* region and PRS.AD models *APOE* separately and subsequently adds this to PRS.no.APOE. AUC Area Under the Curve, ORS Oligogenic risk score, PRS Polygenic risk score, AD Alzheimer's Disease, SNP Single nucleotide polymorphism.

is lower when *APOE* is excluded, but the accuracy remains high using PRS.no.APOE (OR = 95, AUC = 95.7). The ORS.no.APOE accuracy for ε33 carriers drops to AUC = 56.3 with an OR smaller than 1, showing that the prediction is in the wrong direction. Therefore, the oligogenic model is not useful for discrimination between ε33 cases and controls in these data.

Finally, we examined whether the individuals in the extremes are the same across all different PRS methods for both ORS and PRS, with positive and negative extremes considered separately (see pairwise visualisation plot in Supplementary Fig. 4). It can be observed that the greatest number of shared extremes is between PRS(C + T) and PRSice, which was anticipated given the methodological similarities of these approaches. The smallest number of shared identifications is between SBayesR and other methods. Overall, the individuals identified with LDpred-Inf, PRS (C + T), PRSice and PRS-CS overlap considerably, in contrast to LDAK and SBayesR.

It can be seen that there are fewer negative extremes identified by ORS than by PRS in all methods. This is explained by the fact that

ORS is predominantly driven by *APOE*-ε4, with the consequence that ORS is not very good at identifying negative extremes. Additional plots for mapping the top and bottom 5 PRS.no.APOE extremes in ε3 homozygotes across different methods are presented in Supplementary Fig. 5. The individuals with the most extreme PRS in both the positive and negative tails are consistent between PRS(C + T) and PRSice, while the identified extremes may differ substantially across the other different PRS methods.

**Discussion**

Identifying individuals at high and low polygenic risk is very important for further work to understand how genetic risk translates into mechanisms of disease[28]. This might also become very relevant for drug development efforts targeting precise mechanisms of disease, as the PRS scores could be used to select small samples of people in which proof of concept for the treatment can be obtained before testing the drug in larger cohorts. The accuracy of current models precludes the use of PRS in the clinic (too many false positives and false negatives). Results

**Table 3 Number of ORS/PRS extremes in the case-control dataset standardised within the sample and against 1000 Genomes European population.**

| Sample | Risk Score | Tail | In-sample standardisation | | | | Population-based standardisation | | | |
|---|---|---|---|---|---|---|---|---|---|---|
| | | | N case (%) | N controls (%) | OR (95% CI) | AUC | N cases (%) | N controls (%) | OR (95% CI) | AUC |
| All | ORS.full | Positive | 18 (6.6) | 2 (0.7) | 9 (0.4, 207) | 84.2 | 33 (12) | 5 (1.8) | 10 (1, 75) | 74.6 |
| | | Negative | 1 (0.3) | 1 (0.3) | | | 2 (0.7) | 3 (1.1) | | |
| | PRS.full | Positive | 11 (4) | 2 (0.7) | 20 (3, 145) | 81.3 | 19 (7) | 3 (1.1) | 32 (6, 180) | 83.1 |
| | | Negative | 3 (1.1) | 11 (3.9) | | | 3 (1.1) | 15 (5.3) | | |
| | PRS.AD | Positive | 21 (7.7) | 1 (0.3) | 100 (3, 2989) | 84.5 | 33 (12) | 3 (1.1) | 124 (6, 2707) | 88.2 |
| | | Negative | 0 (0) | 3 (1) | | | 0 (0) | 6 (2) | | |
| ε3ε3 | ORS.no. APOE | Positive | 1 (1) | 3 (1.8) | 1.7 (0.1, 38) | 43.8 | 1 (1) | 2 (1.1) | 0.6 (0.03, 14) | 56.3 |
| | | Negative | 1 (1) | 5 (3) | | | 1 (1) | 3 (1.8) | | |
| | PRS.no. APOE | Positive | 4 (4) | 1 (0.6) | 39 (1, 1191) | 99.9 | 7 (7) | 2 (1.1) | 95 (3, 2683) | 95.7 |
| | | Negative | 0 (0) | 6 (3.6) | | | 0 (0) | 10 (6) | | |

Legend: In a case-control dataset the number of cases (N cases) and controls (N controls) at PRS extremes were identified with percentage from total of cases = 271 and total of controls = 278. The prediction accuracy of these extremes was assessed with AUC and OR (95% Confidence Intervals) when standardised (a) using sample mean and SD (b) using mean and SD from 1000 Genomes data. We define PRS extremes as individuals with a score exceeding ± 2 SD from the data mean or population mean. Three models were used for the whole dataset (549 individuals): ORS.full (pT ≤ 1e-5), PRS. full (pT ≤ 0.1) and PRS.AD (pT ≤ 0.1) and two models were used for ε3 homozygote individuals (N = 267 with 100 cases and 167 controls): ORS.no.APOE and PRS.no.APOE. ORS.no.APOE and PRS.no. APOE exclude the *APOE* region and PRS.AD models *APOE* separately and subsequently adds this to PRS.no.APOE.

of this and other studies[18] confirm that identification based on having a PRS above/below a certain threshold provides much better prediction accuracy than attempting to classify all individuals in a dataset. However, there is no agreement in the field about the best way to model risk and to generate risk scores for reliable and accurate identification of high and low-risk individuals. We examined the major factors that need to be considered when calculating PRS for AD and provide recommendations to aid individual selection.

Firstly, we provide ample evidence that AD should be modelled as a polygenic disease. We advocate that risk of AD is not different from other diseases where liability to disease is continuous, and that disease becomes evident after a threshold has been passed (the liability threshold model). In the threshold model liability for a genetic disorder is normally distributed across the population, and polygenic risk scores are a measure of disease liability[29]. While common alleles of small risk identified by genome-wide association study arrays capture between a third and a half of the genetic variance in liability to AD, *APOE*-ε4 alone substantially increases risk for the disorder[28]. Here we show that AD cases with *APOE*-ε4 alleles have a lower burden of common AD risk alleles of small effect than AD cases without ε4 alleles. This implies that *APOE* risk is substantial for the development of the disease with a lesser burden of common risk alleles with small effects (the genetic liability threshold model[30]).

Secondly, a major problem with AD when using PRS to categorise people at risk is the age of the study participants. Allelic variation at the *APOE* locus impacts survival by both altering the age at onset of AD and by increasing risk of other conditions (hyperlipidaemia, atherosclerosis, cardiovascular disease[31–36]), and the frequency of *APOE*-ε4 in the population correspondingly goes down with age[11]. Furthermore, it has been shown that PRS's contribution to dementia (where AD is the most common form of dementia) risk differs with age and *APOE*-ε4 allele status[37]. The effect of PRS (pT ≤ 0.5) is more pronounced in older people[25], and the effect of oligogenic risk scores constructed using SNPs with an association pT ≤ 1e-5 is greater in ε4 homozygotes[17]. In this study, comparing the means of the oligogenic and polygenic risk scores across age groups, we found that following the pattern of *APOE*-ε4 frequency, the ORS decreased with age in cases but was on average higher than in controls (conversely, PRS increases with age in cases, but decreases in controls). A potential

explanation is that *APOE* and most of the GWAS significant SNPs point to genes which are in the same or overlapping pathways[38–42]. This also explains why including the oligogenic scores does not improve the prediction very much compared to *APOE* genotype alone. In addition, since *APOE* is associated with earlier age at onset, these shared pathway genes are likely to be detectable in younger and/or mixed age samples.

As a consequence, the best predictive accuracy was achieved in this and in our earlier study[14] using the regression model with two predictors: *APOE* and PRS.no.APOE (i.e., PRS at pT ≤ 0.1, which excludes the *APOE* region). Our simulation study confirms that if controls are drawn from a general population, and hence are younger than cases, the model where PRS is included as one variable suggests a lower optimal pT, similar to those observed in other studies[16,17,43]. Notably, in this study, the prediction accuracies of the PRS using p-value thresholds of 0.1 and 0.5 were similar (the latter reported in earlier work by us and others[14,15]). The reduction of the optimal pT from 0.5 to 0.1 is likely due to the improved estimation of SNP effect sizes, imputation quality and increased GWAS sample size in the latest GWAS[13] in comparison to the earlier GWAS study[44]. Similar findings have been observed for other polygenic disorders, e.g., in the Schizophrenia and Bipolar datasets of the Psychiatric Genetic Consortium[45]. Note, however, that the optimal p-value threshold may differ when predicting other endophenotypes, for example CSF plasma, imaging biomarkers or other types of dementia[46,47].

Thirdly, comparing six PRS calculation methods, we conclude that the prediction accuracies are very similar, however the individuals' scores differ. The choice of the individuals at the extremes of the PRS distribution were concordant between PRSice, LDpred-inf, PRS-CS and PRS(C + T). There were more differences shown between LDAK and SBayesR. Due to the lack of transparency within the Bayesian approaches, it is difficult to explain why certain individuals are at high polygenic risk whereas others are not, compared to PRS(C + T) where the SNP effect sizes and the LD clumping parameters are traceable. While functionally informed methods may reach higher prediction accuracy in a population, the posterior SNP effect sizes will differ from the true effect sizes if they were obtained from, e.g., multivariate regression[48].

Another interesting and important conclusion of our study is that overlaying a relatively small case/control sample onto the general population results in a much better representation of

risk in the sample. Since case-control samples are enriched for cases as compared to the general population, the PRS distribution of the former is a mixture of two distributions (cases and controls) with distinct means. The PRS distribution for a population sample is likely to have a mean between the means of cases and controls, and a smaller variance (and hence, standard deviation) than that of the combined case-control sample. Standardising the case-control sample to the population will result in the shift of the individual scores in the case-control sample to the positive or negative side of the population mean. This makes the detection of more individuals at high or low risk possible. Increasing the size of the population sample will provide better estimates of the population PRS mean and SD (since the standard errors of these estimates will decrease as $N$ increases). Note that including a larger population sample will proportionally increase the total number of people more than 2SDs from the mean in the joint (population plus case-control) sample, but that this will not necessarily be enriched by the individuals from the case-control sample. For example, the use of the 1000G population is an easy and straight-forward way to obtain this beneficial effect.

Finally, we looked at individuals at the extremes of the PRS distribution and found that both odds ratio and AUC were very high in the whole sample (OR = 124, 95% CI = [6, 2707]) and for the ε3-homozygous individuals (OR = 95, 95% CI = [3, 2683]). The confidence intervals for the odds ratios were of course broad, as the sample size is small when looking at the extremes, but the accuracy remained high. The ORs for the extremes identified by ORS were smaller (OR = 10, 95% CI = [1,75]) and the CIs narrower, suggesting that this model identified a greater number of extremes than the polygenic model, but with poorer accuracy. The oligogenic score was not suitable to identify the extremes in the ε33 individuals with OR = 0.6, misclassifying some high ORS cases as controls and vice versa.

There are a number of limitations in this study. The size of the case control sample used was small, which may have reduced the power and decreased the precision of the accuracy measures used. This can be seen in the broad CIs of the OR for PRS, for example. Secondly, clinical definitions of AD varied across the cohorts which were combined to generate our case-control sample, and age was not recorded uniformly (e.g., age at interview vs age of death). This would make our sample more heterogeneous, thereby possibly also decreasing the power. Lastly, when excluding APOE from the model, we removed the whole APOE locus as this is a region of high LD. In doing so we may have inadvertently excluded SNPs which are independently associated with AD over and above APOE. To ensure generalisability, these results require replication in independent datasets.

In conclusion, we show that for AD the optimal p-value threshold is pT ≤ 0.1, and the PRS calculation should account for the age-related genetic differences in cases and controls either by modelling APOE separately to the PRS or by matching cases and controls for age and APOE status. This approach can be refined when we have a better idea of which genes are contributing to the disease aetiology via aging and which are directly on the pathology pathway. We recommend the use of PRS(C + T) for the calculation of risk scores as this led to the highest accuracy and variance explained in our analyses. To ensure comparability between studies and samples, researchers should standardise PRS against an appropriate population, and use the extremes of the standardised PRS distribution to select individuals at high/low risk of AD.

## Methods

**Data sets and quality control.** The 1000 Genomes (1000 G) Project[49] applied whole genome sequencing to individuals from different populations in order to

compile a detailed resource of common human genetic variation. In this study we only consider individuals from a European population, $N = 503$.

The UK Biobank (UKBB) is a large prospective cohort of approximately 500,000 individuals from the UK containing extensive phenotypic and genotypic data which is still being collected[50]. Participants recruited were aged 39–73 years with a mean age of 56.8. The data here were used under UKBB approval for application (15175: Further defining the genetic architecture of Alzheimer's disease) and contain 443,018 individuals after Quality Control (QC) analysis.

HipSci (Human Induced Pluripotent Stem Cell Initiative)[51] is an initiative which is generating a large, high quality reference panel of human iPSC lines for the research community. These are created from tissue donations from both healthy volunteers and patients from particular rare disease communities. There were 1,228 samples from healthy volunteers available from this study.

ADNI (Alzheimer's Disease Neuroimaging Initiative) is a longitudinal study that was developed for the early detection of AD with the use of clinical, genetic, and imaging data[52]. The data was collected from 900 participants between ages 55 and 90. Initially, participants were followed for 2–3 years with repeated imaging scans and psychometric measurements (ADNI1). The study was subsequently extended with the addition of new participants (ADNI-GO and ADNI2). Longitudinal data contained information on clinical assessments from the first, baseline visit to the latest available visit with mean follow up time approximately 5 years. Genetic data was available for 770 participants who provided written consent.

ROSMAP—Religious Orders Study (ROS) and the Rush Memory and Aging Project (MAP) are both ongoing longitudinal clinical-pathologic cohort studies of aging and AD. Older participants were recruited without dementia and multi-layer data was collected that includes structural and functional neuroimaging, quantitative clinical phenotypes, neuropathologic and neurobiological traits, multi-level omics and genetics[53–55]. 1,196 samples were available to us with genetic information.

MSBB (The Mount Sinai Brain Bank) study generated gene expression, genomic variant, proteomic and neuropathological data from brain specimens. Clinical dementia rating scale (CDR) was conducted for assessment of dementia and cognitive status[56]. 349 samples were available to us with genetic information.

MAYO—Mayo Clinic Brain Bank is a post-mortem cohort that contains neuropathological, genetic, biochemistry, cell biology data. The samples that are used here are described in MAYO eGWAS[57]. 349 samples were available to us with genetic information.

All standard Quality-Control (QC) steps were performed separately in each dataset using PLINK[58], see Supplementary Note 1. The 1000 Genomes Project and UK Biobank were used as population cohorts for PRS standardisation. The HipSci dataset was used as an example of a population cohort for the identification of individuals with high and low AD risk. ADNI, ROSMAP, MSBB, and MAYO were used as part of a combined case-control cohort. To gain more power we combined and harmonised ADNI, ROSMAP, MSBB and MAYO studies, removed overlapping samples that were used in the Kunkle et al GWAS study[13], leaving 271 AD cases and 278 controls with 6,077,045 SNPs for the remaining analysis (see details in Supplementary Table 1 and Supplementary Table 2).

**Primary PRS calculation (C + T).** For the PRS calculation we used the summary statistics from the largest available clinically assessed case-control GWAS study on AD[13] ($N = 63,926$) to generate genetic scores for all participants in the cohorts described above as the weighted sum of the risk alleles. The most commonly used approach for PRS calculation is clumping and thresholding (C + T) where markers most strongly associated with disease are preferentially retained. PRS were generated with the PLINK genetic data analysis toolset[58] for pT ≤ 5e-8, 1e-5, 0.1, 0.5 on LD-clumped SNPs by retaining the SNP with the smallest p-value excluding variants with $r^2 > 0.1$ in a 1000-kb window. Additional PRSs were computed with more stringent $r^2$ thresholds of 0.01 and 0.001 for all p-value thresholds. PRS.no. APOE was calculated excluding the APOE region (chromosome 19:44.4–46.5 Mb) due to the high LD in this region. PRS.AD was calculated as a weighted sum of PRS.no.APOE and APOE(ε2 + ε4), where APOE effects were weighted with effect sizes (B(ε2) = −0.47 and B(ε4) = 1.12) as in Kunkle et al. 2019[13]. Prior to any analyses, all derived scores were adjusted for principal components (PCs) and then standardised (a) within the sample and (b) against population cohorts. For the latter, the dataset was merged with the population data, PCs were derived on the merged data, then the data was standardised using the mean and standard deviation (SD) from the population subsample.

**Other methods of PRS calculation.** We computed PRS using a number of different methods, in particular PRSice, LDpred-inf, PRS-CS, LDAK and SBayesR taking the effect sizes from the Kunkle summary statistics[13]. To maintain a fair comparison, all PRS methods are applied to an identical dataset containing the same set of thresholded SNPs, i.e., ORS (pT ≤ 1e-5) and PRS (pT ≤ 0.1). We also computed PRS using the whole genome data without any prior clumping and thresholding with LDpred-inf, PRS-CS and LDAK, but software issues prevented us from being able to run this with SBayesR. LD was estimated using the case-control dataset for all methods with the exception of PRS-CS, which used the 1000 Genomes data (as this was the only option available in the PRS-CS software). All

methods were otherwise implemented using default options. The PRS generated were standardised against the 1000 Genomes population data.

**Statistical analysis**. The case-control association analysis was performed using logistic regression with the glm() function in R. The prediction accuracy was estimated in terms of (a) area under the receiver operating characteristic curve (AUC) and (b) $R^2$, the proportion of the variance explained by the regression model. The extremes at $\pm 2$ SD were compared in terms of OR with 95% Confidence Intervals (CI), AUC, cases and controls at each tail of the PRS distribution, and pairwise overlap between the extremes for all methods. For the PRS extremes we compare the results of ORS (pT $\leq$ 1e-5) and PRS (pT $\leq$ 0.1), including the PRS.AD model. We used the Haldane correction[59] in instances when cell counts were zero in the $2 \times 2$ contingency table.

**Simulation study**. Independent genotypes were simulated in a sample of 10,000 cases and 10,000 controls. APOE-ε4 allele frequency was set at 0.142 in controls and 0.356 in cases[60]. For simplicity, we assumed that the age of cases is above the late onset (e.g., over 85) but the age of the controls is below the average early onset (e.g., below 60 years). To estimate the number of controls who will develop the disease at 84, we used results from[9], which show a frequency of 91% of ε4ε4 homozygote donors having AD, and a mean age of onset of 68; for ε4 heterozygotes this is 47% and 76 years, and 20% of ε4 non-carriers with onset at an average of 84 years of age. This suggests that there are about 28% hidden or putative cases among the controls. Then we re-simulated ε4 genotypes with slightly reduced allele frequency (f = 0.355) in cases and slightly elevated allele frequency (f = 0.36) for putative controls, so that the joint allele frequency is ~0.356 for the true cases (10,000 + 2,800) and for the 10,000 young population controls matching the distribution of ε4 frequency by age[10]. We set the frequency of ε4 to 0.142 for the remaining controls[13]. ORS.full was calculated based upon 68 LD-clumped SNPs with pT $\leq$ 1e-5, including one of the most significant APOE variants (rs429358), with frequencies and effect sizes as reported in summary statistics from Kunkle et al. 2019[13]. PRS.full was calculated using 10,068 SNPs, where 68 SNPs had effect sizes as described above and the other 10,000 SNPs were simulated with minor allele frequencies uniformly distributed between 0.01 and 0.45 (70%/30% of SNPs with minor/major risk allele) and effect sizes decreasing from OR = 1.005 to 1 and from OR = 1.003 to 1 for the cases and putative controls, respectively; see R script (code availability).

**Reporting summary**. Further information on research design is available in the Nature Research Reporting Summary linked to this article.

## Data availability

Kunkle et al. 2019 summary statistics for the Stage 1 can be obtained from The National Institute on Aging Genetics of Alzheimer's Disease Data Storage Site (NIAGADS)—a NIA/NIH-sanctioned qualified-access data repository, under accession NG00075. UK Biobank data used in this study were available under UK Biobank approval (https://www.ukbiobank.ac.uk/, application 15175). Alzheimer's Disease Neuroimaging Initiative (ADNI) used in this study were available at the database (http://adni.loni.usc.edu/), upon registration and compliance with the data usage agreement. ROSMAP, MSBB, MAYO datasets were available via Synapse platform (https://www.synapse.org: syn3191087, syn10901595, syn6101474, syn10901600, syn3817650, syn10901601) and https://www.radc.rush.edu. 1000 Genomes data are publically available and can be found at http://www.1000genomes.org. All HipSci data were accessed from http://www.hipsci.org (HipSci data access agreement 8759).

## Code availability

The code can be downloaded from https://github.com/DRI-Cardiff/APOE-modelling.

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

## Acknowledgements

We thank the Dementia Research Institute [UKDRI supported by the Medical Research Council (UKDRI-3003), Alzheimer's Research UK, and Alzheimer's Society], Welsh Government, Joint Programming for Neurodegeneration (MRC: MR/T04604X/1), Dementia Platforms UK (MRC: MR/L023784/2), MRC Centre for Neuropsychiatric Genetics and Genomics (MR/L010305/1), The Moondance Foundation, VIB and KU Leuven (Methusalem grant), the European Union (grant no. ERC-834682 CELLPHA-SE_AD), the "Fonds voor Wetenschappelijk Onderzoek", the "Geneeskundige Stichting Koningin Elisabeth", Opening the Future campaign of the Leuven Universitair Fonds, the Belgian Alzheimer Research Foundation, and the Alzheimer's Association USA. B.D.S. is a holder of the Bax-Vanluffelen Chair for Alzheimer's disease. Data used in preparation of this article were obtained from the Alzheimer's Disease Neuroimaging Initiative (ADNI) database (adni.loni.usc.edu). As such, the investigators within the ADNI contributed to the design and implementation of ADNI and/or provided data but did not participate in analysis or writing of this report. A complete listing of ADNI investigatorscan be found at: http://adni.loni.usc.edu/wp-content/uploads/how_to_apply/ADNI_Acknowledgement_List.pdf. Data collection and sharing for this project was funded by the Alzheimer's Disease Neuroimaging Initiative (ADNI) (National Institutes of Health Grant U01 AG024904) and DOD ADNI (Department of Defense award number W81XWH-12-2-0012). ADNI is funded by the National Institute on Aging, the National Institute of Biomedical Imaging and Bioengineering, and through generous contributions from the following: AbbVie, Alzheimer's Association; Alzheimer's Drug Discovery Foundation; Araclon Biotech; BioClinica, Inc.; Biogen; Bristol-Myers Squibb Company; CereSpir, Inc.; Cogstate; Eisai Inc.; Elan Pharmaceuticals, Inc.; Eli Lilly and Company; EuroImmun; F. Hoffmann-La Roche Ltd and its affiliated company Genentech, Inc.; Fujirebio; GE Healthcare; IXICO Ltd.; Janssen Alzheimer Immunotherapy Research & Development, LLC.; Johnson & Johnson Pharmaceutical Research & Development LLC.; Lumosity; Lundbeck; Merck & Co., Inc.; Meso Scale Diagnostics, LLC.; NeuroRx Research; Neurotrack Technologies; Novartis Pharmaceuticals Corporation; Pfizer Inc.; Piramal Imaging; Servier; Takeda Pharmaceutical Company; and Transition Therapeutics. The Canadian Institutes of Health Research is providing funds to support ADNI clinical sites in Canada. Private sector contributions are facilitated by the Foundation for the National Institutes of Health (www.fnih.org). The grantee organization is the Northern California Institute for Research and Education, and the study is coordinated by the Alzheimer's Therapeutic Research Institute at the University of Southern California. ADNI data are disseminated by the Laboratory for Neuro Imaging at the University of Southern California. Data also were provided by the Rush Alzheimer's Disease Center, Rush University Medical Center, Chicago. Data collection was supported through funding by NIA grants P30AG10161, R01AG15819, R01AG17917, R01AG30146, R01AG36836, U01AG32984, U01AG46152, U01AG61356, the Illinois Department of Public Health, and the Translational Genomics Research Institute. ROSMAP data can be requested at https://www.radc.rush.edu. This study makes use of data generated by the HipSci Consortium, funded by The Wellcome Trust and the MRC and UK Biobank Resource, under application number 15175.

## Author contributions

G.L., E.B., J.S.H.—performed statistical analysis and wrote the manuscript; A.S., M.F., J.W.—reviewed and approved the manuscript and added valuable comments; B.d.S., V.E.P.—conceived and designed the study and wrote the manuscript.

## Competing interests

The authors declare no competing interests.
