## [Peer Review File · Nature Communications]

Reviewers' Comments:

Reviewer #1:

Remarks to the Author:

In this manuscript Leonenko and colleagues investigate different methodologies and assumptions used for calculations of polygenic risk scores (PRSs) for AD. Further, they present how to standardize PRSs using a population-based material. In general, I think the manuscript is well-written and very interesting to read. It adds important information within the field of genetics in Alzheimer's disease.

Comments to the authors:

1. Introduction: I would suggest that the aims of the study are presented in a (slightly) more structured way. Now it seems like the aims are presented a bit repetitively, and I do not understand the last part of the last sentence (but maybe that is due to a typo) "...we investigated and PRS standardisation against a population".

2. Introduction: In the section describing the effect of age and APOE e4 status on the PRS contribution to AD (line 117 and onwards) the authors might want to include the novel article of Najar et al. (<https://alz-journals.onlinelibrary.wiley.com/journal/23528729>) showing an effect of PRS for AD on dementia risk only in APOE e4 non-carriers (up to very old age).

3. Methods (section "Primary PRS calculation (P+T)): Why do the authors prefer to use LD-pruning instead of LD-clumping? I am also very interested to see what happens with the results if the r^2 threshold (for LD) is reduced to a stricter level (such as 0.01, or even 0.001)?

4. Methods (section "Other methods of PRS calculation"): The authors write that "...all PRS methods are applied to an identical dataset containing the same set of thresholded SNPs ($p_T \leq 5e-8$, $1e-5$, 0.1, 0.5). However, from what I can see only the $1e-5$ and the 0.1 level results are presented in the results section (Figure 2).

5. Results: Throughout the results section there are wording that suits better into a discussion part; some examples: Line 415 "...which is not surprising", line 417: "The results become much more interesting...", line 456: "We advise using PRS(P+T or PRSice...". The authors should preferably avoid interpreting the results in this way in the results section.

6. Discussion: I just want to comment that I think the discussion is very well-written!! The only thing I would suggest is that the authors perhaps could comment on that the optimal p-value threshold to use for constructing an AD-PRS might differ depending on the outcome that is investigated (for instance when studying relations between AD-PRS and endophenotypes, such as CSF/plasma/imaging biomarkers).

Minor comments:

Define the meaning of P and T in PRS(P+T).

The authors should tell which LD-reference genome that was used for PRS constructions using the PRSice software.

On line 252 (methods section "Simulation study") there seems to be words missing in the sentence beginning with "ORS.full...". Perhaps it should be "ORS.full was calculated...?"

At line 305 in the results, section "Optimal p-value threshold": The sentence "To allow for..." is very long and difficult to read (and therefore to understand).

Reviewer #2:

Remarks to the Author:

1. The manuscript requires editing for English language and grammar.
2. The authors speak to personalized prevention and intervention therapies, though I don't know that many such strategies exist for AD.
3. I would argue with the assertion that AD cases are relatively easy to detect clinically (without imaging). The authors also don't make explicit how the ability to diagnose AD follows from knowledge about risk factors.
4. I found the Introduction to be rather unwieldy. It is unclear how the various tangents will have to do with the goals of the study. The authors can considerably tighten to better motivate their research. Perhaps some of the material would be better suited for the Discussion.
5. When describing each of the contributing data sources, it might be helpful to have some indication of the purpose of each contribution. And can the authors make clearer how the final sample size was achieved? I also worry about population effects given the number of contributing data sources.
6. Did the authors adjust for any covariates in their PRS analyses? Age perhaps?
7. How is it simple to simulate different age distributions for cases and controls?
8. Can the authors use the Discussion to summarize the major contribution(s) of their work? Something that ties their analyses together?
9. Might the Discussion address any limitations? E.g., generalizability?
10. My final comment – a reiteration – is that I found that the story wasn't cohesive. I admit that I lost the thread on many occasions and struggled to follow the messaging. Perhaps what I'm suggesting is that the paper would feel cleaner if the authors were trying to do fewer things.

Reviewer #1 (Remarks to the Author):

In this manuscript Leonenko and colleagues investigate different methodologies and assumptions used for calculations of polygenic risk scores (PRSs) for AD. Further, they present how to standardize PRSs using a population-based material. In general, I think the manuscript is well-written and very interesting to read. It adds important information within the field of genetics in Alzheimer's disease.

Comments to the authors:

1. Introduction: I would suggest that the aims of the study are presented in a (slightly) more structured way. Now it seems like the aims are presented a bit repetitively, and I do not understand the last part of the last sentence (but maybe that is due to a typo) "...we investigated and PRS standardisation against a population".

Many thanks for your comment, we have re-structured and edited the introduction section. We improved the focus of the paper as the robust identification of individuals with high/low AD risk based on their PRS score.

In order to do this we set to investigate the following questions:

- 1) how to include effect of APOE in PRS model*
- 2) what is the optimal p-value threshold for SNP selection*
- 3) comparison of different PRS approaches both in terms of prediction accuracy and identification of high/low risk individuals*
- 4) how to standardise PRS scores to ensure comparability between studies.*

2. Introduction: In the section describing the effect of age and APOE e4 status on the PRS contribution to AD (line 117 and onwards) the authors might want to include the novel article of Najjar et al. (<https://alz-journals.onlinelibrary.wiley.com/journal/23528729>) showing an effect of PRS for AD on dementia risk only in APOE e4 non-carriers (up to very old age).

Thank you for highlighting this paper, we have added the reference in discussion section, line 321.

3. Methods (section "Primary PRS calculation (P+T)": Why do the authors prefer to use LD-pruning instead of LD-clumping? I am also very interested to see what happens with the results if the r^2 threshold (for LD) is reduced to a stricter level (such as 0.01, or even 0.001)?

Thank you for your comment, we did in fact use LD clumping and have added a sentence to section 4.2 clarifying that we used clumping and thresholding when we refer to pruning and thresholding (P+T), line 457.

Additionally, we have now re-calculated PRS for $r^2=0.01$ and $r^2=0.001$, the results for prediction accuracy can be seen in Supplementary Table 1. The prediction accuracy is

slightly reduced when using more stringent r^2 thresholds, and this reduction is greater with a more stringent r^2 threshold. We have added text to Result section 3.1, line 159 and 463.

4. Methods (section “Other methods of PRS calculation”): The authors write that “...all PRS methods are applied to an identical dataset containing the same set of thresholded SNPs ($p_T \leq 5e-8, 1e-5, 0.1, 0.5$). However, from what I can see only the $1e-5$ and the 0.1 level results are presented in the results section (Figure 2).

Thank you for highlighting this, initially we investigated 4 p-value thresholds ($p_T \leq 5e-8, 1e-5, 0.1, 0.5$) in terms of precision accuracy. Since we observed little difference in precision between $p_T < 5e-8$ and $1e-5$, and between $p_T < 0.1$ and $p_T < 0.5$, see Table 2, we only presented the comparisons for the two thresholds $p_T < 1e-5$ and 0.1 . We have appropriately corrected the text in Methods 4.3, line 477.

5. Results: Throughout the results section there are wording that suits better into a discussion part; some examples: Line 415 “...which is not surprising”, line 417: “The results become much more interesting...”, line 456: “We advise using PRS(P+T or PRSice...”. The authors should preferably avoid interpreting the results in this way in the results section.

Thank you for this comment, we have amended the results section 2.4 and discussion, so that results are interpreted in the discussion.

6. Discussion: I just want to comment that I think the discussion is very well-written!! The only thing I would suggest is that the authors perhaps could comment on that the optimal p-value threshold to use for constructing an AD-PRS might differ depending on the outcome that is investigated (for instance when studying relations between AD-PRS and endophenotypes, such as CSF/plasma/imaging biomarkers).

Thank you for your kind comment, we have added a sentence into the Discussion Section, see line 343.

Minor comments:

Define the meaning of P and T in PRS(P+T).

We have added this to the text, see line 457.

The authors should tell which LD-reference genome that was used for PRS constructions using the PRSice software.

LD was estimated from the case-control dataset where possible for all PRS methods (the only exception was PRS-CS where you have to use 1000 Genomes). We have edited the text to clarify this, line 486.

On line 252 (methods section “Simulation study”) there seems to be words missing in the sentence beginning with “ORS.full...”. Perhaps it should be “ORS.full was calculated...”?

Thanks, we have corrected this.

At line 305 in the results, section “Optimal p-value threshold”: The sentence “To allow for...” is very long and difficult to read (and therefore to understand).

Thank you, we have removed this sentence, since it repeated information from the Method section.

Reviewer #2 (Remarks to the Author):

1. The manuscript requires editing for English language and grammar.

Thank you for your comment, we have corrected English and grammar and re-structured some of the sentences to improve readability.

2. The authors speak to personalized prevention and intervention therapies, though I don't know that many such strategies exist for AD.

We have removed that phase.

3. I would argue with the assertion that AD cases are relatively easy to detect clinically (without imaging). The authors also don't make explicit how the ability to diagnose AD follows from knowledge about risk factors.

We agree with the reviewer that it is not easy to detect AD without imaging. Our original sentence was unclear, we wanted to comment on the fact that in the majority of GWAS studies, AD cases have a clinical (or maybe pathological) diagnosis but controls are not age-matched and may develop AD in the future. We have edited the text to clarify our meaning and also discuss dementia incidence based on knowledge of other associated risk factors, line 87.

4. I found the Introduction to be rather unwieldy. It is unclear how the various tangents will have to do with the goals of the study. The authors can considerably tighten to better motivate their research. Perhaps some of the material would be better suited for the Discussion.

Thank you for your comment, we have re-structured and edited the abstract and introduction sections. We have focused our aims on the robust identification of individuals with high/low AD risk based on their PRS score.

In order to do this we set to investigate the following questions:

1) how to include effect of APOE in PRS model

2) what is the optimal p-value threshold for SNP selection

3) comparison of different PRS approaches both in terms of prediction accuracy and identification of individuals

4) how to standardise PRS in order to compare scores and selection of individuals between studies.

5. When describing each of the contributing data sources, it might be helpful to have some indication of the purpose of each contribution. And can the authors make clearer how the final sample size was achieved? I also worry about population effects given the number of contributing data sources.

In Section 4.1 we have added information on how each dataset was used in the analysis, line 440. Detailed information about these datasets, their QC and harmonization on both genetic and phenotypic levels can be found in Supplementary Table 4 and 5, and Supplementary Section 1. All the datasets were individually QCed.

In order to improve power of the case-control study we have combined the 4 datasets available to us (ADNI, ROSMAP, MAYO and MSBB). Polygenic risk scores were adjusted for principal components computed from the merged set, in order to account for different population effects across cohorts. These studies are well established for studying AD and have some cognitive or pathological information and are very well known in AD field.

The final sample size was achieved by combining all genetic information together (N=770 for ADNI, N=1196 for ROSMAP, N=349 for MAYO and N=349 for MSBB), performing QC, extracting clinically defined cases and controls in each cohort, and finally removing individuals that were originally included in Kunkle et al.

The number of cases and controls for each cohort contributing to the analysis can be found in Supplementary table 5.

Additionally we have included this as a limitation of our study in Discussion section, line 381.

6. Did the authors adjust for any covariates in their PRS analyses? Age perhaps?

Thank you for your comment, all PRSs were adjusted for PCs, see Supplementary Section 1. We have not adjusted for age because the purpose of the study was to identify individuals with high/low AD risk using genetic information only. However, we have additionally investigated PRS/ORS scores and APOE genotype changes with age (see Figure 1). We also discuss how differing age distributions in cases and controls may lead to the disagreement in the field regarding the optimal p-value threshold for SNP selection for the construction of PRS, line 177.

7. How is it simple to simulate different age distributions for cases and controls?

Since we did not use age as covariate in our analysis, the actual values of age were not needed to be simulated. We have simulated the APOE index SNP frequencies, reflecting the e4 allele frequencies distributions in two age groups assuming the mean age in

controls ~54 years and in cases ~79 years (as e.g. a sample of AD cases and population controls described in Moreno-Grau et al (2019)).

e4 frequencies:

- a) 0.355 for hidden cases, i.e. current controls who will get AD later in life: (28% of all controls, N=2,800)
- b) 0.360 in current cases (N=10,000)
- c) 0.142 in controls, who will not get AD (72% of all controls, N=7,200)

The weighed average of a) and b) is $0.356 = (2800*0.355 + 10000*0.36)/12800$ corresponding to the frequencies in cases and controls and the OR=3.326 as reported in (Kunkle et al (2019)).

For the PRS, additional independent (reflecting the “LD-pruning”) SNPs were generated with allele frequencies, reflecting the effect size (OR) (reflecting the “thresholding”) 10,000 cases and 10,000 controls without accounting for age.

The details of the simulation study is described in Section 4.5 and script is available to run (<https://github.com/DRI-Cardiff/APOE-modelling>).

8. Can the authors use the Discussion to summarize the major contribution(s) of their work? Something that ties their analyses together?

Thank you for your comment, as mentioned previously, we have refocused the aims of our study and restructured the discussion to highlight the major contribution which is the robust detection of individuals at high/low AD risk.

9. Might the Discussion address any limitations? E.g., generalizability?

Thank you for highlighting this, we have added a limitations section to the discussion, line 381.

10. My final comment – a reiteration – is that I found that the story wasn’t cohesive. I admit that I lost the thread on many occasions and struggled to follow the messaging. Perhaps what I’m suggesting is that the paper would feel cleaner if the authors were trying to do fewer things.

Thank you again for your useful comments, we have focused the aims of the paper and hope that the revised version is more cohesive.

Reviewers' Comments:

Reviewer #1:

Remarks to the Author:

I think the manuscript has improved after the revisions made by the authors. Overall, I am also happy with the way my comments/questions have been answered, I just have two additional (minor) to follow up on that:

1. I realize that I was a bit vague in my question on pruning and clumping. I meant that I do not understand why the authors prefer to use the word pruning when it is apparent that clumping has been performed? In contrast to clumping, pruning does not take the p value of the SNPs into account, so why not use the expression "clumping and thresholding" (C+T) instead (throughout the manuscript)? (Based on my previous comment 3).

2. Ref 32 that has now been added in the discussion is about AD-PRS in relation to dementia (where the majority are AD-cases, but not all), and not in relation to AD only (which the authors now state). Maybe the authors want to clarify this. (based on my previous comment 2).

Reviewer #2:

Remarks to the Author:

The authors did a good job addressing my comments. Because the manuscript is now much more readable, I am able to offer a bit more feedback:

1. The Introduction is much improved. Just a couple outstanding questions:

a. What does "vascular aspects" mean? And how are they modifiable? I wouldn't think that they could be considered lifestyle factors?

b. Why would decreased dementia incidence result in controls enriched for future AD cases? If changes in lifestyle have truly led to reduced dementia, then wouldn't the effects be seen in controls as well? (I understand the logic that some portion of controls that are younger than cases are likely to develop AD, but don't understand the argument surrounding lifestyle.)

2. In the Results paragraph describing the simulation, " $p \leq 1e-510$ " seems to have a typo. Assuming that the threshold is in fact $1e5-10$, why so stringent?

3. The last sentence of Results Section 2.1 is incomplete.

4. Though the Introduction offers some information about different methods to calculate PRS, each should be defined when first mentioned in the Results. E.g., the reader does not necessarily know what "LDAK" means at first mention.

a. Reading further into the Methods, I'm wondering whether they were originally written to be placed before the Results. The Methods include various clarifications that might be useful in the Results (e.g., what the "case-control dataset" is).

5. Did the authors consider standardization relative to sample controls only?

6. The results regarding PRS extremes would benefit from specification of a denominator.

Reviewer #1 (Remarks to the Author):

I think the manuscript has improved after the revisions made by the authors. Overall, I am also happy with the way my comments/questions have been answered, I just have two additional (minor) to follow up on that.

1. I realize that I was a bit vague in my question on pruning and clumping. I meant that I do not understand why the authors prefer to use the word pruning when it is apparent that clumping has been performed? In contrast to clumping, pruning does not take the p value of the SNPs into account, so why not use the expression “clumping and thresholding” (C+T) instead (throughout the manuscript)? (Based on my previous comment 3).

Thank you for the clarification of your previous comment. We now use “clumping and thresholding” (C+T) throughout the manuscript.

2. Ref 32 that has now been added in the discussion is about AD-PRS in relation to dementia (where the majority are AD-cases, but not all), and not in relation to AD only (which the authors now state). Maybe the authors want to clarify this. (based on my previous comment 2).

We have commented the paper correctly, line 331.

Reviewer #2 (Remarks to the Author):

The authors did a good job addressing my comments. Because the manuscript is now much more readable, I am able to offer a bit more feedback:

1. The Introduction is much improved. Just a couple outstanding questions:
a. What does “vascular aspects” mean? And how are they modifiable? I wouldn't think that they could be considered lifestyle factors?

We refer to the work in reference [2], where the authors discuss modifiable risk factors for dementia. These include vascular risk factors (e.g., hypertension, obesity, high blood glucose), which are modifiable by exercise and diet. We have now corrected the sentence to make it clearer:

About 35% of life-time risk of dementia is modifiable by factors such as education, nutrition, health care and social deprivation², including a better management of vascular risk factors with their prevalence decreasing over time.

b. Why would decreased dementia incidence result in controls enriched for future AD cases? If changes in lifestyle have truly led to reduced dementia, then wouldn't the effects be seen in controls as well? (I understand the logic that some portion of controls that are younger than cases are likely to develop AD, but don't understand the argument surrounding lifestyle.

Thank you, very good point. The assumption is not that the prevalence of dementia is decreasing, it is actually increasing, but the age at which dementia manifests itself is moving. That is clear from Framingham's and many other studies. We have corrected this statement, making it clearer, and cited additional studies.

The effects of better management of these risk factors manifests in a delay in age at onset of the disease as noticed in many studies³⁻⁵. However, with increased lifespan the prevalence of the disease still goes up⁶. If controls are enrolled from the population and/or are younger than cases, then a proportion of them will develop AD at a later time⁷. Due to potential delay of the age at onset, even age-matched control samples are likely to encompass future AD cases who are yet to show symptoms.

2. In the Results paragraph describing the simulation, “ $p \leq 1e-510$ ” seems to have a typo. Assuming that the threshold is in fact $1e5-10$, why so stringent?

Thank you for pointing this out. It was not a typo but reference. In order to avoid confusion, we have placed it different place in the sentence.

In the simulation we have used p-value threshold $< 1e-5$ that corresponds to ORS model and additionally simulated 10,000 SNPs that represents PRS model.

3. The last sentence of Results Section 2.1 is incomplete.

Thanks for pointing this out, we have completed the sentence.

4. Though the Introduction offers some information about different methods to calculate PRS, each should be defined when first mentioned in the Results. E.g., the reader does not necessarily know what “LDAK” means at first mention.

Thanks for the point. We have moved the description of PRS approaches to the Results section.

a. Reading further into the Methods, I'm wondering whether they were originally written to be placed before the Results. The Methods include various clarifications that might be useful in the Results (e.g., what the “case-control dataset” is).

In our original submitted version, the Methods section was before the Results. But in order to follow the resubmission guidelines of the Journal, we now have to move Results after Introduction and put Methods after Discussion. We now have defined case-control dataset in the Results Section.

5. Did the authors consider standardization relative to sample controls only?

We think that standardisation against a population cohort provides the most reliable way to compare PRS distribution parameters from different studies and to

choose PRS extremes of a study of interest. The standardisation relative to sample controls only, may not be suitable for different study designs (i.e., case-control study, only case study, only control study or population cohort with unknown disease status). Even the definition of “controls” often differs between studies. Of course, the population samples are also not perfect, but they are often publicly available (e.g. 1000genomes) and can be used by many researchers in the world, thus making the selection of PRS extremes comparable between different studies.

6. The results regarding PRS extremes would benefit from specification of a denominator.

This comment was not very clear to us. We assumed that the reviewer meant the percentage of correctly and not correctly identified cases and controls from the total numbers of cases and controls in the study. We have added this information to Table 3.

Reviewers' Comments:

Reviewer #2:

Remarks to the Author:

The authors adequately addressed my comments.